# Scaffolding and completing genome assemblies in real-time with nanopore sequencing

Minh Duc Cao[1,*], Son Hoang Nguyen[1,*], Devika Ganesamoorthy[1], Alysha G. Elliott[1], Matthew A. Cooper[1] & Lachlan J.M. Coin[1]

Third generation sequencing technologies provide the opportunity to improve genome assemblies by generating long reads spanning most repeat sequences. However, current analysis methods require substantial amounts of sequence data and computational resources to overcome the high error rates. Furthermore, they can only perform analysis after sequencing has completed, resulting in either over-sequencing, or in a low quality assembly due to under-sequencing. Here we present npScarf, which can scaffold and complete short read assemblies while the long read sequencing run is in progress. It reports assembly metrics in real-time so the sequencing run can be terminated once an assembly of sufficient quality is obtained. In assembling four bacterial and one eukaryotic genomes, we show that npScarf can construct more complete and accurate assemblies while requiring less sequencing data and computational resources than existing methods. Our approach offers a time- and resource-effective strategy for completing short read assemblies.

[1] Institute for Molecular Bioscience, University of Queensland, St Lucia, Brisbane, Queensland 4072 Australia. * These authors contributed equally to this work. Correspondence and requests for materials should be addressed to M.D.C. (email: m.cao1@uq.edu.au) or to L.J.M.C. (email: l.coin@imb.uq.edu.au).

High-throughput sequencing technology has transformed genomics research over the last decade with the ability to sequence the whole genome of virtually any organism on the planet. Most sequencing projects to date employ short read technology and hence cannot unambiguously resolve the repetitive sequences that are present abundantly in most genomes. As a result, assemblies are fragmented into large numbers of contigs and the positions of repeat sequences in the genome cannot be determined. These repeat sequences often play important biological roles; for example, they mediate the lateral transfer of genes between bacterial species via pathogenicity islands and plasmids. Analysing these regions is thus essential to determine key characteristics such as anti-microbial resistance (AMR) or to identify highly pathogenic variants of many bacterial species[1].

Long read sequencing technologies, for example Pacific Biosciences' (PacBio) and Oxford Nanopore MinION sequencing, allow users to generate reads spanning most repetitive sequences, which can be used to close gaps in fragmented assemblies. A key innovation of the MinION nanopore sequencing device is that it measures the changes in electrical current as a single-stranded molecule of DNA passes through the nanopore and uses the signal to determine the nucleotide sequence of the DNA strand[2–4]. As such, the raw data of a read can be retrieved and analysed as soon as it is generated, while sequencing of other reads is still in progress. This offers the opportunity to obtain analysis results as soon as sufficient data are generated, upon which sequencing can be terminated or used for other experiments.

Several algorithms have been developed to utilize long reads for genome assembly. de novo assemblers such as the hierarchical genome assembly process[5] and nanocorrect/nanopolish[6] can assemble a complete bacterial genome using only long read sequencing data. However, because of the high error rates in these sequencing technologies, this de novo approach requires substantial amounts of sequencing data and extensive computational resources, mainly for polishing the genome assembly. Hybrid assemblers, which combine error-prone long reads with highly accurate and cheaper short read sequence data, provide a more economical and efficient alternative for building complete genomes. They can be classified into three categories. de novo methods such as Canu[7] and Miniasm[8] employ fast approximate approaches to assemble a skeleton of the genome using long reads. The skeleton, often as erroneous as the raw reads, is then polished with high quality short reads. On the other hand, tools in the error-correction category (for example, PBcR[9], Nanocorr[10] and NaS[11]) correct long reads with high quality short reads before assembling the genome with the corrected long reads. Finally, the scaffolding methods (SPAdes-hybrid[1,12], SSPACE-LongRead[13,14] and LINKS[15]) use long reads to scaffold and fill in gaps in the assemblies from short read sequencing.

While these tools are reported to assemble high quality bacterial genomes[16,17], they have not made use of the real-time sequencing potential of the MinION; assembly of a genome can only be performed after the sequencing is complete. This can lead to over-sequencing, which incurs extra cost and time; or under-sequencing resulting in a low-quality assembly. Here, we present npScarf, the first hybrid assembler that can scaffold and complete fragmented short read assemblies with sequence data streaming from the MinION while sequencing is still in progress. In effect, npScarf can fully utilize a sequence read within minutes of it being generated. Furthermore, it continuously reports assembly quality during the experiment so that users can terminate the sequencing when an assembly of sufficient quality and completeness is obtained. We show that our method can generate more accurate and more complete genomes than

existing tools, while requiring less nanopore sequencing data and computational resources. As such, npScarf can be used to efficiently control MinION sequencing in completing existing short read assemblies and in hybrid assembly projects. More importantly, npScarf can facilitate the real-time analysis of positioning genomic sequences for time critical applications such as in AMR investigation. We show that the npScarf can rapidly and accurately reconstruct genomic islands carrying AMR genes that are fragmented in short read assemblies. It can also identify AMR genes encoded in plasmids. These are among the main analyses to understand the acquisition of AMR in pathogenic bacteria.

## Results

**Algorithm overview**. The genomes of most organisms contain an abundance of repeat sequences that are longer than the read length limit (300 bps) of Illumina sequencing platforms[18]. In assembling a genome using this technology, these repeat sequences cannot be distinguished and hence are often collapsed into contigs, leaving gaps in the genome assembly. To complete the assembly, npScarf first determines the multiplicity of each contig, thereby identifying contigs representing non-repetitive sequences (called unique contigs). It then scaffolds and fills in gaps in the assembly in a streaming fashion (Fig. 1). Upon receiving a long read from the MinION, npScarf immediately aligns it to the unique contigs. Reads aligned to two unique contigs form a bridge connecting the two contigs. Gradually, the unique contigs are joined to form the scaffold of the genome, while the repetitive contigs are used to fill in the gaps in the scaffold. The details of the algorithms are presented in 'Methods' section.

**Completing bacterial assemblies**. We assessed the performance of our algorithm for its ability to scaffold and complete the Illumina assemblies of two bacterial *Klebsiella pneumoniae* strains, ATCC BAA-2146 (New Delhi mellalo-beta-lactamase (NDM-1) positive) and ATCC 13883 (type strain). We first sequenced the genomes of these strains with the Illumina MiSeq platform to 250-fold coverage and assembled them with SPAdes[12] (see 'Methods' section). This resulted in assemblies of 90 and 69 contigs that were 500 bps or longer, respectively. The N50 statistics of the two assemblies were 288 and 302 Kb, respectively. We then sequenced the two strains with Oxford Nanopore MinION using chemistry R7. For ATCC BAA-2146, we obtained 185 Mb of sequencing data (~33-fold coverage of the genome), of which 27 Mb were two-directional (2D) reads. The run for strain ATCC 13883 yielded only 13.5 Mb of sequencing data (~2.4-fold coverage). We re-sequenced this strain with the improved chemistry R7.3. By combining sequencing data from both experiments for this strain, we obtained a total of 100 Mb (~18-fold coverage) data, including 22.5 Mb of 2D reads. The quality of the data, described in ref. 19, was broadly similar to that reported by other MinION users[1,20,21].

As the pipeline described here was developed after we performed the MinION sequencing runs, we tested our streaming analysis by re-running the base-calling using the Metrichor service. Sequence reads in fast5 format were written to disk, and were instantaneously picked up and streamed to the pipeline by npReader[22]. In essence, the scaffolding pipeline received sequence data in fastq format in a streaming fashion as if a MinION run was in progress. During analysis, the pipeline continuously reported the assemblies' statistics (the numbers of contigs and the N50 statistic), allowing us to track the completeness of the assembly, as well as the number of circular sequences in the genome. This is especially important for the analysis of

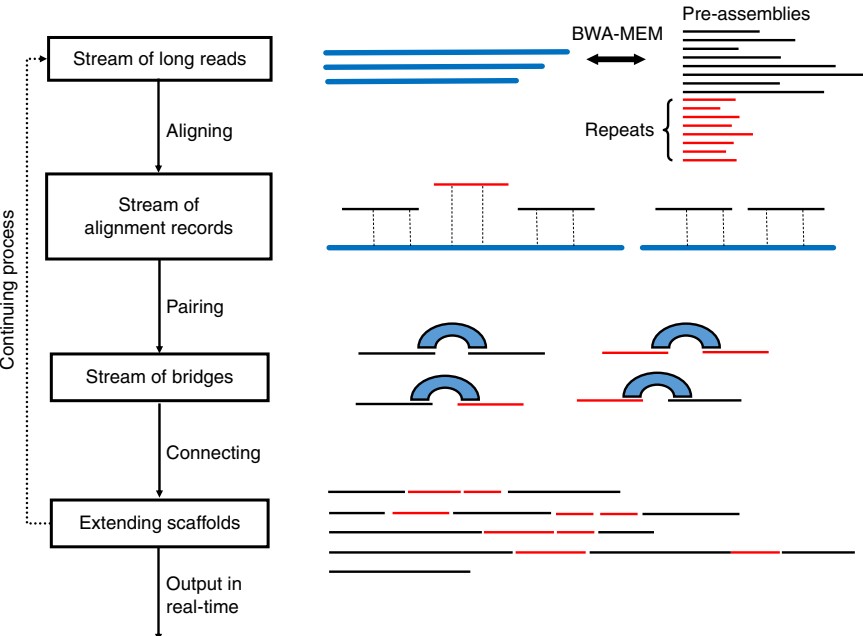

**Figure 1 | Workflow of the real-time algorithm.** Stream of long reads are aligned to the existing contigs to create alignment records. Bridges connecting contigs are formed, and are used for extending scaffolds. These steps are performed in a streaming fashion.

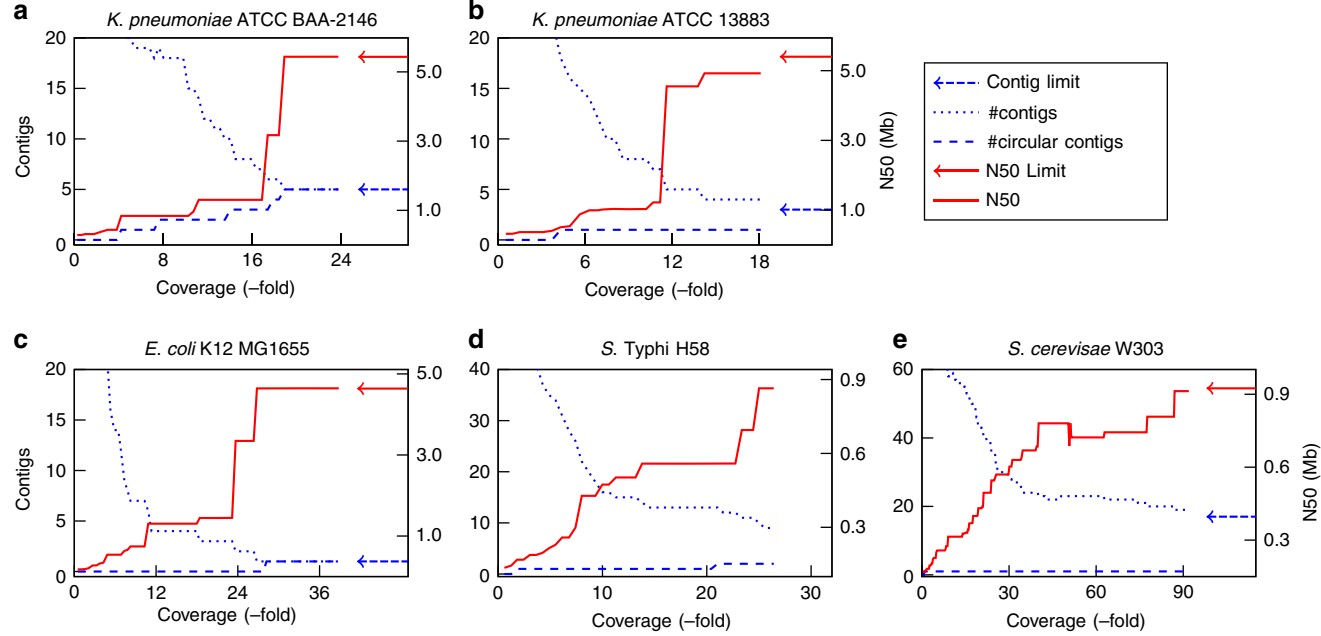

**Figure 2 | Assembly statistics during real-time scaffolding.** The plots show N50 statistics, number of contigs, and number of circular contigs against the amount of nanopore sequencing data for (**a**) *K. pneumoniae* ATCC BAA-2146, (**b**) *K. pneumoniae* ATCC 13883, (**c**) *E. coli*, (**d**) *S.* Typhi H58, and (**e**) *S. cerevisae* W303.

bacterial genomes where chromosomes and plasmids are usually circular. To validate the resulting assemblies, we compared them with the reference genomes of these strains obtained from NCBI (GenBank Accessions GCA_000364385.2 and GCA_000742135.1). We also ascertained the predicted plasmids in these assemblies by looking for the existence of plasmid origins of replication sequences from the PlasmidFinder database[23].

Figure 2a,b present the progress of assembly completion against the coverage of MinION data during scaffolding. As

expected, N50 statistics increased and the number of contigs decreased with more MinION data. For *K. pneumoniae* ATCC BAA-2146, we found that our algorithm required only 20-fold coverage of sequence data (<120 Mb) to complete the genome, reducing the assembly to the limit of five contigs (one chromosome and four plasmids). Those five contigs were circularized, indicating completeness. We found these five contigs to be in agreement with the complete genome assembly of the strain, previously sequenced with PacBio and Illumina[24] (see Table 1 and Supplementary Fig. 1).

**Table 1 | Comparison between npScarf's assemblies and the reference genomes of two _K. pneumoniae_ strains.**

| | npScarf assemblies | | Reference sequences | | |
|---|---|---|---|---|---|
| **Name** | **Size (bp)** | **Plasmid _ORI_** | **Accession** | **Size (bp)** | **Plasmid _ORI_** |
| _K. pneumoniae_ ATCC BAA-2146 | | | | | |
| Contig 1* | 5,437,518 | — | CP006659.1* | 5,435,369 | — |
| Contig 2* | 141,026 | IncA/C2 | CP006661.1* | 140,825 | IncA/C2 |
| Contig 3* | 118,278 | IncFIB(K); IncFII(K) | CP006663.1* | 117,755 | IncFIB(K); IncFII(K) |
| Contig 4* | 85,233 | IncR; IncFIA(HII) | CP006662.1* | 85,164 | IncR; IncFIA(HII) |
| Contig 5* | 2,015 | ColRNAI | CP006660.1* | 2,014 | ColRNAI |
| | | | | | |
| _K. pneumoniae_ ATCC 13883 | | | | | |
| Contig 1 | 4,923,970 | — | KN046818.1 | 5,284,261 | — |
| Contig 2 | 372,214 | — | | | |
| Contig 3 | 139,480 | IncFIA(HII); IncFIB(K) | KN046820.1 | 95,930 | IncFIA(HII); IncFIB(K) |
| | | | KN046821.1 | 42,420 | — |
| Contig 4* | 119,388 | ColRNAI; IncFII(pCoo); pSM22 | KN046819.1 | 106,842 | IncFII(pCoo); pSM22 |
| | | | KN046822.1 | 16,331 | — |

*Circular sequences.

**Figure 3 | Structure of a pathogenic island from _K. pneumoniae_ ATCC BAA-2146.** The island harbors three antibiotic resistance genes _strep_, _sul1_ and _ebr_, flanked by mobility genes integrase (_int_), inverstase (_hin_), DNA replication (_dnaC_) and IS (IS26 and IS6100). The island was fragmented into 10 contigs in the Illumina assembly, and was completely resolved with 65 Mb out of the total of 185 Mb of nanopore sequence data.

With 18-fold coverage of the MinION data for _K. pneumoniae_ ATCC 13883, the assembly was improved to four contigs, in which one was reported to be circular (Contig 4). These contigs were aligned to the reference genome for this strain, which contained 16 contigs in five scaffolds. We found Contig 1 and Contig 2 from the npScarf's assembly were aligned to the reference scaffold KN046818.1, while Contig 3 and Contig 4 were aligned to two reference scaffolds (see Table 1 and Supplementary Fig. 2). The alignments contained forward and reverse matches. The breakpoints of these matches corresponded to the contig joins in the reference scaffolds, indicating the incorrect orientation of contigs in the reference scaffolds. The reference scaffold KN046818.1 was 5.2 Mb in size, suggesting this scaffold was the chromosome and was fragmented into two contigs in the npScarf assembly. In examining this chromosomal sequence, we found the two contigs to be separated by an rRNA operon of 7 Kb in length. BLAST search revealed the structure of this operon with rRNA 5 S, 23 S and 16 S as the main components. This rRNA operon sequence was also found to be present at five other loci in the genome, which were all resolved. However, no long MinION read was found to align to this particular position, possibly because of the low yield of this data set, which caused the chromosome sequence to be fragmented. We anticipate this could be resolved with more nanopore sequencing data. Contig 3 (139 kb) and Contig 4 (119 kb) contained several origin of replication sequences (see Table 1), suggesting they were plasmid sequences; Contig 4 was also reported to be a circular sequence. In Contig 4, we noticed an extra plasmid origin of replication sequence (ColRNAI) that was not found in the reference genome

(see Table 1). In examining the position of ColRNAI, we found it was in one of the gaps in the reference scaffold, hence not reported in the reference assembly.

**Real-time analysis for positional information**. The ability to complete genome assemblies in a streaming fashion also enables real-time analyses that rely on positional information. Such analyses include identifying genes encoded in bacterial genomic islands and plasmids. These functional regions in bacterial genomes can be horizontally transferred between organisms, which is one of the main mechanisms for acquiring AMR in pathogenic bacteria. Here we demonstrate these analyses on the multi-drug resistant _K. pneumoniae_ ATCC BAA-2146 strain.

Before scaffolding the Illumina assembly of the sample, we annotated the assembly using Prokka[25] to identify the positions of genes and insertion sequences (IS) in the assembly. Bacterial genomic islands are genomic regions longer than 8 Kb containing certain classes of genes such as AMR genes. In addition, they often carry mobility genes such as transposase, integrase and IS[26]. These sequences generally appear multiple times in the genomes (repetitive sequences), causing genomic islands fragmented in the short read assembly. We ran Islander[27] and PHAST[28] on the Illumina assembly, which together detected six genomic islands. In the annotation, we also found 28 IS; 14 of these were within 3 Kb of the contig ends, suggesting that any genomic islands flanked by these IS were fragmented. During scaffolding of the assembly with nanopore sequencing data, npScarf constructed four additional genomic islands, which were not previously reported by Islander and PHAST (data not shown).

**Table 2 | Timeline of determining plasmid-encoded antibiotic resistance genes.**

| Data required | Gene ID | NCBI ref | Antibiotic resistance | Plasmid evidence |
|---|---|---|---|---|
| 10 Mb | blaTEM-1B | JF910132 | Penicillins, some cephalosporins | IncR;IncFIA(HI1) |
| | strB | M96392 | Streptomycin | IncR;IncFIA(HI1) |
| | strA | AF321551 | Streptomycin | IncR;IncFIA(HI1) |
| | sul2 | GQ421466 | Sulfonamides | IncR;IncFIA(HI1) |
| 14 Mb | aac6Ib | M21682 | Tobramycin, amikacin, netilmicin, sisomicin | IncR;IncFIA(HI1) |
| 21 Mb | mphA | D16251 | Erythromycin | IncFIB(K);IncFII(K) |
| | tetA | AJ517790 | Tetracyclines | IncFIB(K);IncFII(K) |
| | QnrB7 | EU043311 | Quinolones | IncR;IncFIA(HI1) |
| 29 Mb | dfrA14 | DQ388123 | Trimethoprim | IncR;IncFIA(HI1) |
| 46 Mb | blaNDM-1 | FN396876 | Penicillins, cephalosporins, carbapenems | IncA/C2 |
| 51 Mb | rmtC | AB194779 | Aminoglycosides (include gentamicin, kanamycin) | IncA/C2 |
| 78 Mb | sul1 | AY224185 | Sulphonamide | IncA/C2 |
| | aac6Ib_1 | M21682 | Tobramycin, amikacin, netilmicin, sisomicin | IncA/C2 |
| | blaCMY-6 | AJ011293 | Penicillins, some cephalosporins | IncA/C2 |
| 83 Mb | blaSHV-11 | GQ407109 | Penicillins, some cephalosporins | IncR;IncFIA(HI1) |
| | aac6Ib | M21682 | Tobramycin, amikacin, netilmicin, sisomicin | IncR;IncFIA(HI1) |
| | blaOXA-1 | J02967 | Penicillins | IncR;IncFIA(HI1) |
| | aac3-IIa | X51534 | Gentamicin, tobramycin, netilmicin, sisomicin | IncR;IncFIA(HI1) |

All 10 genomic islands were precisely in agreement with the analysis of the PacBio assembly by ref. 24. Figure 3 presents the structure of one genomic island, namely Kpn23SapB, and the timeline of its reconstruction. This genomic island harbored three AMR genes, namely *aadA* (mediates resistance to streptomycin and spectinomycin), *sul1* (sulfonamides) and *ebr* (ethidium bromide and quaternary ammonium). This genomic island also carried two copies of the insertion sequence IS26, which flanked the AMR genes, and a copy of the insertion sequence IS6100. The presence of these repetitive sequences caused the island to be fragmented into 10 contigs in the Illumina assembly; the three resistance genes were in two different contigs. npScarf required 64.59 Mb of data (14-fold coverage of the genome) to report the full structure of the island (Fig. 3).

For real-time detection of plasmid-encoded genes, we identified plasmid origin of replication sequences from the Illumina assembly using the PlasmidFinder database[23]. Contigs containing a plasmid origin of replication sequence were considered to be part of a plasmid. Essentially, only 166 genes contained within these contigs could be ascertained as plasmid-encoded genes from the Illumina sequencing of the *K. pneumoniae* ATCC BAA-2146 strain. During scaffolding of the Illumina assembly, once a contig was added to a plasmid, npScarf reported genes in the contig as plasmid-encode genes. The amount of long-read sequence data required to assign each gene to a plasmid is presented in the Supplementary Spreadsheet.

With the Illumina assembly, we identified 27 AMR genes, but none was in a contig containing a plasmid origin of replication sequence. As such, whether any of these genes were carried by a plasmid could only be ascertained with long reads. Table 2 presents the time-line of such determination. In particular, we confirmed 18 AMR genes as plasmid-encoded with 83 Mb (∼14-fold coverage) of nanopore sequencing data. In addition, as all four plasmids were circularized and complete with 103 Mb (∼18-fold coverage) of data, we could confidently conclude that only these 18 AMR genes were plasmid-encoded, even before the completion of the full genome assembly and the sequencing run.

**Comparison with other methods**. We compared the performance of our algorithm against existing methods that were reported to build assemblies with nanopore sequencing.

In addition to the two samples presented above, we sourced three other samples reported in the literature including (i) an *Escherichia coli* K12 MG1655 strain sequenced to 67-fold coverage with a nanopore R7.3 flowcell and standard library preparation[29]; (ii) a *Salmonella enterica* serovar Typhi (*S*. Typhi) haplotype, H58 (ref. 1) sequenced to 27-fold and (iii) a *Saccharomyces cerevisiae* W303 genome (196-fold)[10]. Note that the coverage reported here was from all base-called data (including both 1D and 2D reads). Of the methods selected for comparison, SPAdes-hybrid[12], SSPACE-LongRead[13], LINKS[15] and npScarf were scaffolders, whereas Nanocorr[10] and NaS[11] belonged to the error correction category. We assembled the Illumina data of these samples using SPAdes[12] before running the scaffolding methods with nanopore data. SPAdes-hybrid was run by incorporating nanopore data into the assembly (with -nanopore option). The two error correction tools Nanocorr and NaS were run on the nanopore sequencing data using about 50-fold coverage of Illumina data, as suggested by authors of the respective publications. The corrected reads were then assembled using Celera Assembler[30]. We observed that the quality of the assemblies produced by Celera Assembler were highly sensitive to the parameters specified in the specification file. We therefore ran Celera Assembler for each data set on three specification files provided by the authors of NaS and Nanocorr, and report here the most complete assembly obtained. We also ran two popular *de novo* assembly methods, Canu[7] and Miniasm[8] on these data sets. These methods necessitated a polishing step using Pilon[31].

We evaluated the assemblies in terms both of completeness and accuracy. The completeness of an assembly was assessed by N50 statistics and the number of contigs that were longer than 500 bp. To examine the accuracy of an assembly, we compared it with the closest reference genome of the samples in NCBI (see 'Methods' section) to obtain the number of misassemblies, mismatches and short indels. During the test, we recorded the CPU times required by these pipelines to produce the assemblies. Run times for the scaffolder methods included times for running SPAdes and for scaffolding, while those for NaS and Nanocorr included correction time and Celera Assembler time. The times reported for the *de novo* methods included that for polishing using Pilon. Table 3 presents the comparison metrics of all assemblies as reported by Quast[32], as well as their run times.

**Table 3 | Comparison of assemblies produced by npScarf and the comparative methods.**

| Method | Assembly size (Mb) | #Contigs (≥ 500 bp) | N50 (Kp) | Mis-assemblies | Error (per 100 Kb) | Run times (CPU hrs)* | | |
|---|---|---|---|---|---|---|---|---|
| *K. pneumoniae* ATCC BAA-2146. Nanopore data: 33 × coverage | | | | | | | | |
| SPAdes | 5.70 | 90 | 288 | 0 | 4.72 | 15.63 | | |
| SPAdes-Hybrid | 5.75 | 17 | 3,076 | 1 | 6.61 | 16.07 | | |
| SPAdes + SSPACE | 5.74 | 53 | 400 | 4 | 12.73 | 15.63 | + | 2.3 |
| SPAdes + LINK | 5.74 | 31 | 554 | 5 | 16.05 | 15.63 | + | 4.03 |
| SPAdes + npScarf (rt) | 5.78 | 5 | 5,438 | 0 | 20.00 | 15.63 | + | 1.6 |
| SPAdes + npScarf (b) | 5.78 | 5 | 5,438 | 0 | 22.76 | 15.63 | + | 0.84 |
| NaS + CA | 5.89 | 29 | 345 | 15 | 18.89 | 324.35 | + | 3.49 |
| Nanocorr + CA | 5.68 | 68 | 139 | 8 | 141.32 | 312.64 | + | 1.37 |
| Canu + Pilon | 0 | — | — | — | — | — | | — |
| Miniasm + Pilon | 0 | — | — | — | — | — | | — |
| *K. pneumoniae* ATCC 13883. Nanopore data: 18 × coverage | | | | | | | | |
| SPAdes | 5.51 | 69 | 302 | 5 | 6.22 | 16.95 | | |
| SPAdes-Hybrid | 5.54 | 15 | 729 | 19 | 8.02 | 16.97 | | |
| SPAdes + SSPACE | 5.55 | 36 | 685 | 13 | 12.39 | 16.95 | + | 1.48 |
| SPAdes + LINK | 5.55 | 17 | 1,527 | 18 | 16.12 | 16.95 | + | 1.12 |
| SPAdes + npScarf (rt) | 5.55 | 4 | 4,924 | 21 | 10.84 | 16.95 | + | 0.52 |
| SPAdes + npScarf (b) | 5.55 | 4 | 4,924 | 21 | 10.26 | 16.95 | + | 0.45 |
| NaS + CA | 5.46 | 38 | 394 | 36 | 10.24 | 192.78 | + | 6.92 |
| Nanocorr + CA | 5.02 | 60 | 148 | 16 | 118.34 | 161.33 | + | 2.6 |
| Canu + Pilon | 0.04 | 4 | 12 | 4 | 10.40 | 0.53 | + | 0.46 |
| Miniasm + Pilon | 0.03 | 3 | 13 | 1 | 14.12 | 0.00 | + | 0.26 |
| *E. coli* K12 MG1655. Nanopore data: 67 × coverage | | | | | | | | |
| SPAdes | 4.61 | 114 | 176 | 0 | 3.51 | 4.38 | | |
| SPAdes-Hybrid | 4.67 | 42 | 4,643 | 2 | 1.21 | 4.76 | | |
| SPAdes + SSPACE | 4.66 | 59 | 3,155 | 1 | 29.26 | 4.38 | + | 3.42 |
| SPAdes + LINK | 4.66 | 50 | 3,318 | 2 | 36.19 | 4.38 | + | 4.03 |
| SPAdes + npScarf (rt) | 4.64 | 1 | 4,644 | 2 | 13.08 | 4.38 | + | 2.43 |
| SPAdes + npScarf (b) | 4.64 | 1 | 4,646 | 2 | 11.72 | 4.38 | + | 1.91 |
| NaS + CA | 4.87 | 21 | 874 | 19 | 10.60 | 807.19 | + | 6.77 |
| Nanocorr + CA | 4.66 | 2 | 4,650 | 6 | 10.41 | 213.68 | + | 8.49 |
| Canu + Pilon | 0.11 | 9 | 14 | 0 | 13.90 | 0.79 | + | 0.28 |
| Miniasm + Pilon | 1.91 | 85 | 23 | 1 | 595.61 | 0.04 | + | 1.24 |
| *S.* Typhi H58. Nanopore data: 26 × coverage | | | | | | | | |
| SPAdes | 4.84 | 89 | 107 | 7 | 39.05 | 1.86 | | |
| SPAdes-Hybrid | 4.88 | 27 | 443 | 12 | 55.46 | 2.06 | | |
| SPAdes + SSPACE | 4.88 | 34 | 358 | 10 | 59.39 | 1.86 | + | 1.55 |
| SPAdes + LINK | 4.86 | 20 | 473 | 13 | 66.65 | 1.86 | + | 1.28 |
| SPAdes + npScarf (rt) | 4.87 | 9 | 864 | 18 | 53.86 | 1.86 | + | 0.93 |
| SPAdes + npScarf (b) | 4.86 | 8 | 864 | 16 | 52.01 | 1.86 | + | 0.47 |
| NaS + CA | 4.97 | 54 | 212 | 17 | 58.87 | 248.32 | + | 7.21 |
| Nanocorr + CA | 2.98 | 95 | 37 | 9 | 973.63 | 199.85 | + | 0.94 |
| Canu + Pilon | 0 | — | — | — | — | — | | — |
| Miniasm + Pilon | 0.02 | 2 | 14 | 0 | 10.96 | 0.01 | + | 0.26 |
| *S. cerevisae* W303. Nanopore data: 196 × coverage | | | | | | | | |
| SPAdes | 11.82 | 364 | 155 | 29 | 124.10 | 20.54 | | |
| SPAdes-Hybrid | 12.06 | 240 | 346 | 68 | 158.13 | 67.81 | | |
| SPAdes + SSPACE | 13.39 | 263 | 392 | 89 | 136.66 | 20.54 | + | 31.54 |
| SPAdes + LINK | 12.09 | 161 | 580 | 83 | 143.04 | 20.54 | + | 26.97 |
| SPAdes + npScarf (rt) | 12.00 | 19 | 913 | 82 | 141.93 | 20.54 | + | 21.28 |
| SPAdes + npScarf (b) | 11.90 | 17 | 924 | 79 | 141.01 | 20.54 | + | 18.84 |
| NaS + CA | 12.76 | 121 | 155 | 123 | 140.08 | 9811.88 | + | 140.69 |
| Nanocorr + CA | 13.48 | 108 | 600 | 133 | 197.00 | 7208.08 | + | 272.86 |
| Canu + Pilon | 12.31 | 43 | 497 | 81 | 229.08 | 599.36 | + | 58.5 |
| Miniasm + Pilon | 11.79 | 51 | 391 | 41 | 1400.82 | 0.27 | + | 30.27 |

*Where two tools are used, running times are presented separately.

We ran npScarf in real-time mode, in which nanopore sequencing data are streamed to the pipeline in the exact order they were generated. This allowed us to assess the completeness of the assemblies against the amount of data generated. Figure 2 shows the progress of completing the assemblies for all five samples. As mentioned previously, npScarf produced complete and near-complete assemblies for the two *K. pneumoniae* samples (Fig. 2a,b) with under 20-fold coverage of nanopore data. For the *E. coli* K12 MG1655 sample, npScarf required less than 30-fold coverage of nanopore data to complete the genome

assembly with one circular contig. npScarf also reduced the *S.* Typhi assembly to only nine contigs (N50 = 864 kb), which was significantly better than the assembly reported by[1] from the same data (34 contigs, N50 = 319 kbs).

As for the *S. cerevisae* W303 genome, which contains 16 nuclear chromosomes and one mitochondrial chromosome, npScarf generated an assembly of 19 contigs (N50 = 913 Kb); substantially fewer than the 108 contigs (N50 = 600 Kb) generated by the next best method (Nanocorr, see Table 3). We noticed a drop in N50 statistics at the point where about 50-fold coverage of nanopore data were received (Fig. 2e). This was because npScarf encountered contradicting bridges and hence broke the assembly at the lowest scoring bridge in lieu of a higher scoring one. The N50 was then improved to reach the N50 of 913 Kb with 90-fold coverage of nanopore sequencing; the assembly did not change with more data (90-fold to 196-fold). We examined the assembly by comparing that with the reference genome of *S. cerevisae* strain S288C. One of the contigs (Contig 17, length = 81 Kb) was reported to be circular, which was completely aligned to the mitochondrial chromosome of the reference genome. Ten chromosomes (II, IV, V, VII, IX, X, XI, XIII, XV and XVI) were completely assembled into individual contigs, and three chromosomes (I, III and VIII) were assembled into two contigs per chromosome (See Supplementary Fig. 3). We found a misassembly that joined chromosome IV and the start of chromosome XIV into Contig 10. The end of chromosome XIV was also joined with chromosome XII into Contig 2. These misassemblies essentially fused these three chromosomes into two contigs. We found these mis-assemblies were due to the presence of interspersed repeat elements which are known for being problematic in assembly analysis[18]. The assemblies produced by Canu and Miniasm also presented several mis-assemblies fusing different chromosomes together, emphasizing the challenges posed by interspersed repeats in assembling complex genomes (See Supplementary Figs 4 and 5).

We reran npScarf on the data sets in batch mode, in which the scaffolding was performed with the complete data set. We found that all five assemblies were more complete than in real-time mode. In particular, the *S. cerevisae* W303 assembly was further reduced to 17 contigs as chromosomes I and VIII were resolved into individual contigs (data not shown). In this assembly, 12 out of 17 chromosomes were completely recovered to one contig, one chromosome (XIII) was fragmented into two contigs and three chromosomes were fused into two contigs due to misassemblies.

In all data sets, npScarf consistently produced the most complete assemblies, while its accuracy was among the best. It was the only method that was able to completely resolve the *K. pneumoniae* ATCC BAA-2146 genome (five contigs, N50 of 5.4 Mb) with no misassembly, requiring only 20-fold coverage of nanopore data; the second most completed assembly (produced by SPAdes Hybrid) contained 17 contigs and had the N50 of only 3.1 Mb despite using 33-fold coverage of nanopore sequence data. On the well studied *E. coli* K12 MG1655 strain sample where LINK, NaS and Nanocorr were reported to resolve the whole genome with a larger data set (147-fold coverage)[15], none of these methods could produce the same result on the 67-fold coverage data set we tested. On the other hand, npScarf was able to reconstruct the genome into one circular contig with as little as 30-fold coverage of the data. On the *S.* Typhi data set, npScarf produced assemblies with nine contigs in real-time mode, and with eight contigs in batch mode (N50 = 864 Kb), significantly better than assemblies from other methods (over 20 contigs). We observed that the *de novo* methods, Canu and Miniasm failed to construct a skeleton for the these bacterial genomes (either no output or only

a few small sequences produced), possibly due to the low coverage of these data sets.

The *S. cerevisae* W303 assembly produced by npScarf was near complete and N50 statistics reached the theoretical limit of 924 Kb. Note that npScarf obtained these results from only less than half of the data set (95-fold coverage). On the whole data set (196-fold coverage), Canu and Miniasm produced assemblies of 43 contigs (N50 = 496 Kb) and 51 contigs (N50 = 391 Kb), respectively. These assemblies contained more than twice as many contigs as the results from npScarf. The second most complete assembly in terms of N50 statistic was produced by Nanocorr (N50 = 600 Kb) which was significantly lower than that from npScarf.

We observed that the scaffolding methods and the *de novo* methods were much faster than the error correction counterparts. Both NaS and Nanocorr required the alignment of the short reads to the long reads, which were computationally expensive. On the other hand, the scaffolding pipelines required 20 CPU hours or less to build an assembly from short reads, and from a few hours to around 30 h to scaffold the assembly with long reads. As Canu and Miniasm did not produce a decent assembly for the bacterial data sets, we only include them in the comparison for the *S. cerevisae* data set. Miniasm was the fastest on this data set, requiring only 0.27 CPU-hours to assemble and over 30 CPU-hours to polish the genome. Apart from SPAdes-Hybrid, which performed scaffolding as part of short read assembly, npScarf was the fastest among other scaffolders and consistently required less scaffolding time. Note that the times reported in Table 3 were for processing the entire nanopore data set, whereas npScarf could be terminated early once a desirable assembly was obtained. We observed that npScarf required only 2 GB of memory for scaffolding the bacterial data sets, and 4 GB for the *S. cerevisae* data set, which can be easily installed on a laptop computer. A summary of memory usage of other tools was presented in Supplementary Table 1.

## Discussion

The development of high-throughput long read sequencing technologies such as Pacific Biosciences and Oxford Nanopore Technology has opened up opportunities for resolving repetitive sequences to assemble complete genomes and to improve existing genome assemblies. However, the relatively high error rates of these technologies pose a challenge to the accurate assembly of genome sequences. An obvious solution is to combine long and erroneous reads with more accurate and cheaper short read data for assembling genomes[9,33]. One such approach is to perform *de novo* assembly of long reads to generate a skeleton of the genome, and error correct the skeleton with accurate short reads[7,8]. Alternatively, erroneous long reads are corrected[9–11,33] before being assembled with classical assemblers designed for long and accurate reads such as Celera Assembler[30]. These approaches usually require large amounts of long read data. Hybrid assemblers in the scaffolding class harness long spanning reads to guide the extension of contigs in the draft genome assemblies. For example, SSPACE-LongRead[13] and Cerulean[34] rely on the alignment of long reads to the assembly graph to determine the adjacent contigs. LINKS[15] uses a k-mer approach, which further improves the running time with a small sacrifice of accuracy. Overall, hybrid-assembly methods, especially those in the scaffolding category, provide economical genome finishing pipelines that can produce high-quality genome assemblies from small amounts of long read data on modest computing equipment. npScarf is similar to these mentioned scaffolders in the sense that it aligns the long reads to the contigs to build a scaffold of the genome. However, our method estimates the copy

number of each contig in the genome and constructs the scaffold from non-repetitive contigs, while the repetitive contigs are used to fill the gaps in the scaffold. Consequently, we demonstrated that npScarf is capable of generating more complete and accurate assemblies than the competitors, while requiring much less data.

To date, there is no prominent assembler that takes advantage of the real-time feature from nanopore sequencing. Nanopore technology allows one to terminate a run and wash the flowcell for subsequent runs without compromising sequencing yield and quality. The ability to analyse data on the fly and to stop a sequencing run when sufficient data are generated plays a critical role to control resources necessary for a single experiment.

One of the main contributions of our algorithm is that it can process data streaming from the sequencer and report the current status of the analysis in real-time. Our pipeline still relies on a base-caller, and the introduction of fast real-time base-callers such as Nanocall[35] and DeepNano[36] helps to reduce the latency. The current pipeline processes a sequence read within minutes of it finishing traversing the pore, rather than as the read is actually passing through the pore, and as such is real-time at the temporal resolution of minutes, but not at the millisecond level required to update with the addition of each base. However, this temporal resolution is sufficient to allow our pipeline to answer the biological problems at hand at the earliest possible time, and while sequencing is still in progress. Investigators can also assess the progress of the analysis, and terminate the sequencing once an assembly of sufficient quality and completeness is obtained. This enables the generation of sufficient data necessary for the analysis to guarantee the experimental outcomes and, at the same time, avoids costly over-sequencing. While our pipeline still requires short read data which cannot be generated in real-time with current technology, it offers a strategy to minimize the generation of the more expensive long read data.

The real-time function to complete genomic sequences opens the possibility of *in situ* biological analyses[19]. Certain biological markers of interest may be identified from short read assemblies, but their positions in the genome could only be determined by completing the genome assembly with long reads. We have shown that npScarf can facilitate such analyses in real-time by demonstrating the identification of AMR genes encoded in plasmids and pathogenicity islands.

## Methods

### Determining unique contigs.
Before scaffolding a fragmented short read genome assembly, npScarf determines the multiplicity of each contig in the assembly by comparing short read sequencing coverage of the contig to that of the whole genome. Coverage information is often included in the sequences assembled by most tools, such as SPAdes[12] and Velvet[37], or can otherwise be obtained from the mapping of short reads to the assembly. An reasonable estimate for depth coverage of the genome is that of the largest contig. npScarf however leverages this to the normalized average coverage of the largest contigs so long as their depth coverage does not deviate from the estimated genome depth coverage. More formally, let $depth_i$ and $len_i$, respectively, represent the sequencing depth (coverage) and length of contig $i$, where contigs are sorted in decreasing order in length. Let $depth_g$ represent the estimated coverage of the whole genome. npScarf first initializes $depth_g$ to that of the largest contig:

$$depth_g^1 = depth_1 \qquad (1)$$

It then iteratively updates the estimate

$$depth_g^i = \frac{\sum_i depth_i \times len_i}{\sum_i len_i} \qquad (2)$$

and terminates the process when the depth coverage of the next contig greater than a threshold

$$\frac{depth_i}{depth_g^{i-1}} > \theta \qquad (3)$$

npScarf set the threshold $\theta$ to 1.5. In our experience, the statistic is stable with up to 20 of the largest contigs longer than 20 Kb, which are most likely unique contigs in bacterial genomes[38]. We hence also add these into the condition for termination. The multiplicity of contig $i$ ($mul_i$) is determined by

$$mul_i = \frac{depth_i}{depth_g} \qquad (4)$$

npScarf considers a contig unique if its multiplicity is less than $\theta$.

### Bridging unique contigs and filling gaps with repetitive contigs.
npScarf next builds the backbone of the genome from the unique contigs. It identifies the long reads that are aligned to two unique contigs, thereby establishing the relative position (that is, distance and orientation) of these contigs. To minimize the effect of false positives that can arise from aligning noisy long reads, npScarf groups reads that consistently support a particular relative position into a bridge and assigns the bridge a score based on the number of supporting reads and the alignment quality of these reads. When two unique contigs are connected by a bridge, they are merged into one larger unique contig. npScarf uses a greedy strategy based on Kruskal's algorithm[39], which merges contigs from the highest scoring bridges. In the newly created contig, the gap is temporarily filled with the consensus sequence of the reads forming the bridge. npScarf then identifies repetitive contigs that are aligned to this consensus sequence, and uses these contigs to fill in the gap.

### Real-time processing.
To support real-time analysis of nanopore sequencing, the previously described algorithm can be augmented to process long read data directly from a stream (See Fig. 1). In this mode, npScarf employs a mapping method that supports streaming processing such as BWA-MEM[40] to align a small number of long reads to the existing assembly as they arrive. This block-wise processing allows npScarf to make use of information from a small batch of reads sequenced within a short period of time (within minutes). If a read is aligned to two unique contigs, it is added to the bridge connecting the two contigs. Once the bridge reaches a pre-defined scoring threshold, the two contigs are merged and the gap is filled as above. In case this merging contradicts with the existing assembly (for example, if the relative distance and/or orientation implied by the bridge is inconsistent with those of previously used bridges) npScarf revisits the previous bridges, breaks the smallest scoring contradicting bridge and uses the current bridge instead. The algorithm hence gradually improves the completeness and the quality of the assembly as more data are received.

### Bacterial cultures and DNA extraction.
Bacterial strains *K. pneumoniae* ATCC BAA-2146 (NDM-1 positive) and ATCC 13883 (type strain) were obtained from American Type Culture Collection (ATCC, USA). Bacterial cultures were grown overnight from a single colony at 37 °C with shaking (180 r.p.m.). Whole-cell DNA was extracted from the cultures using the DNeasy Blood and Tissue Kit (QIAGEN, Cat #69504) according to the bacterial DNA extraction protocol with modified enzymatic lysis pre-treatment.

### Illumina sequencing and assembly.
Library preparation was performed using the NexteraXT DNA Sample preparation kit (Illumina), as recommended by the manufacturer. Libraries were sequenced on the MiSeq instrument (Illumina) with 300 bp paired end sequencing, to a coverage of over 250-fold.

### MinION sequencing.
Library preparation was performed using the Genomic DNA Sequencing kit (Oxford Nanopore), according to the manufacturer's instructions. For the R7 MinION Flow Cells SQK-MAP-002 sequencing kit was used and for R7.3 MinION Flow Cells SQK-MAP-003 were used, according to the manufacturer's instructions. A new MinION Flow Cell (R7 or R7.3) was used for each sequencing run. The library was loaded onto the MinION Flow Cell and the Genomic DNA 48-hour sequencing protocol was initiated using MinKNOW software.

### Data collection.
MinION data for the *E. coli* K12 MG1655 sample[20] were downloaded from the European Nucleotide Archive (ENA) with accession number ERP007108. We used the data from the chemistry R7.3 run (67-fold coverage of the genome from run accession ERR637419) rather than the chemistry R7 reported in work by[10,11,15]. Illumina MiSeq sequencing data for the sample were also obtained from ENA (assession number ERR654977). Data from both Illumina and MinION sequencing of the *S.* Typhi strain[1] were collected from ENA accession number ERP008615. The *S. cerevisae* W303 sequencing data were provided by ref. 10 from the website http://schatzlab.cshl.edu/data/nanocorr/.

### Data processing.
Read data from Illumina sequencing were trimmed with *trimmomatic* V0.32 (ref. 41) and subsequently assembled using SPAdes V3.5 (ref. 12). SPAdes was run with the recommended parameters (-k 21,33,55,77,99,127 -careful). SPAdes-Hybrid was run with the inclusion of the -nanopore option. SSPACE and LINKS were run on the original SPAdes' assemblies. For SSPACE, we used the parameters reported to work with MinION reads in ref. 14 (-i 70 -a 1500 -g -5000). In the case of LINKS, a script was adapted from the example run for *E. coli* K12 MG1655 sample to allow 30 iterations of the algorithms being executed for each data

set. NaS and Nanocorr were applied to correct nanopore data from the maximum of 50-fold coverage of Illumina data. The corrected long reads were assembled using Celera Assembler version 8.3 with the configuration files provided by the respective publication. Canu was run with the recommended parameter for nanopore data (-nanopore-raw). Miniasm were run with its default parameter. The short reads were aligned to Canu's and Miniasm's assemblies by BWA-MEM, and Pilon was run on the alignments to polish the assembly sequences.

The Illumina assembly of *K. pneumoniae* ATCC BAA-2146 was annotated using Prokka (version 1.12-beta) with the recommended parameters for a *K. pneumoniae* strain. AMR genes from the assembly were identified using the ResFinder database[42]. Plasmid origin of replication sequences in both *K. pneumoniae* assemblies were identified by uploading the assembly to the PlasmidFinder database[23].

In real-time analysis, npScarf (from japsa version 1.6-06b) aligned incoming long reads using BWA-MEM[40] with the parameters -k11-W20-r10-A1-B1-O1-E1-L0-a-Y-K10000. The -K10000 parameter allowed alignments to be streamed to the scaffolding algorithm after several reads were aligned.

**Comparative metrics.** The assemblies produced by the mentioned methods were evaluated using Quast (V3.2) to compare with the respective reference sequences. The number of contigs, N50 statistics and the number of misassemblies were as per Quast reports. The error rates were computed from sum of the number of mismatches and the indel length. The CPU time for each pipeline was measured using the Linux time command (/usr/bin/time -v); the sum of user time and system time was reported. When a pipeline was distributed across a computing cluster, its CPU time was the sum of that across all jobs.

**Data availability.** Sequencing data for the two *K. pneumoniae* samples were deposited to the ENA. The accession numbers for the MinION sequencing data are ERR1474979 and ERR1474981, and that for the MiSeq sequencing data are ERR1474547 and ERR1474549. The software presented in this article and its documentation is publicly available at GitHub https://github.com/mdcao/npScarf.

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

## Acknowledgements

MAC is an National Health and Medical Research Council Principal Research Fellow (APP1059354). LC is an Australian Research Council Future Fellow (FT110100972). The research is supported by funding from the National Health and Medical Research Council (APP1052303) as well as funding from the Institute for Molecular Bioscience Centre for Superbugs Solutions (610246).

## Author contributions

M.D.C. and L.C. conceived the study. S.N., M.D.C. and L.C. designed and implemented the algorithm. A.E. performed the bacterial cultures and DNA extractions. D.G. performed the MinION sequencing and Illumina sequencing. S.N. and M.D.C. performed the analysis and wrote the first draft of the manuscript. All authors contributed to editing the final manuscript.

## Additional information

**Competing financial interests:** M.C. is a participant of Oxford Nanopore's MinION Access Programme (MAP) and received the MinION device, MinION Flow Cells and Oxford Nanopore Sequencing Kits in return for an early access fee deposit. M.D.C. received travel and accommodation expenses to speak at an Oxford Nanopore-organised conference. None of the authors have any commercial or financial interest in Oxford Nanopore Technologies Ltd.

**Publisher's note**: 

