## [Peer Review File · Nature Communications]

Reviewers' comments:

Reviewer #1 (Remarks to the Author):

I have read Cao and co-workers' manuscript on npScarf with much interest. It describes an assembly scaffolding and finishing tool for hybrid assembly of Illumina and nanopore sequencing data. The stated motivation of the method is to provide a means for real time assembly algorithm that can be used in the field, and inform its user on when to stop a sequencing experiment.

Although the method provides competitive performance in comparison to present state-of-the-art, in my opinion it fails to deliver on its stated goals.

First, because it builds on assembled Illumina sequencing data, and because that step has to be performed offline, the primary use case of npScarf cannot be a field study.

Second, the authors have the misconception that the MinION instrument returns its output one read at a time. Unfortunately (for the authors, but fortunately for the users) this is not the case. The instrument has many nanopores operating at the same time. So the actual data streaming happens as an asynchronous time series from multiple pores.

Having said that, I think the study has merit if the authors are willing to reframe the target application of their tool. It performs better than the comparators on the datasets they have tested, in terms of contiguity, quality and run time. (They should also include memory comparisons in their report.)

Of particular interest was, how the algorithm recovered from misassembled bridges. As the authors have demonstrated (though not explicitly stressed) as the target genome size and complexity increases, the assembly problem gets more challenging. As a result, using more data may help reduce misassemblies. This is reflected in their comparison of real time and batch assemblies of the *S. cerevisiae* dataset. My question is, can the algorithm recover from misassemblies in the batch mode. That is, if the algorithm were run in the batch mode with several partitions of the dataset (say, two or three), can the later batch runs correct for the earlier wrong decisions?

I find the way the method determines unique contigs rather simplistic, and lacks proper statistical treatment. Coverage depth of a contig depends on several factors, including its length and GC content. In general, the longer a contig, the closer its average fold coverage to the overall genome fold coverage, and I do realize that the authors are aware of that (they are considering "up to 20 of the largest contigs longer than 20 Kb".) Of course they may provide a better justification for these thresholds, or making them run time parameters, to help address this issue to a certain extent. However it would not alleviate the problem in the way they apply an inferred cut-off across the board, as shorter contigs will have wider fluctuations in their coverage profiles.

I think, adding some logic to determine the completeness of a bacterial genome/plasmid assembly through key features is a great idea. Though it needs to be better explained how it is integrated in the assembly algorithm, including how to invoke it during a run.

Reviewer #2 (Remarks to the Author):

This is a good paper and well laid out. I am particularly impressed by (a) the variety of datasets that

the authors use, including generating their own and (b) the length of accurate scaffolding this approach takes.

However I have one major concern which must be addressed. The authors make much of their real time aspect of scaffolding. However real time algorithms and methods are only really valid if the entire workflow can be done in real time and stopped; as this method is a hybrid method with a batched based MiSeq run, it really is not at all practical as a real time method. In theory there is a marginal saving in ONT time, but as few people directly reuse a flow cell immediately, this is not really useful.

For me this should be recast as a prelude to future "all real time" methods rather than implying it could do something radical now. ie - I think it is fine to show the plots and the concept, but the authors should be up front by saying this is only really useful when the entire workflow can be stopped real time.

Minor comments

Line 23 "The major advancement..." - I would say "A..." or the "the key component of the technology..."

I prefer "less data" to "fewer data"

You need to provide the accession number of the MiSeq.

Reviewer #3 (Remarks to the Author):

I read the manuscript by Cao et al., titled "Scaffolding and completing genome assemblies in real-time with nanopore sequencing" with great interest. The authors describe a method dedicated to de novo genome assembly of small genomes by combining data generated using the Oxford Nanopore and the Illumina sequencing technologies. Moreover, authors take advantage of the "real-time" capability of the MinION device, and they propose to improve an existing assembly in real-time. The manuscript particularly well described the impact of sequencing coverage on the assembly contiguity. The npScarf software is of broad interest for the nanopore users. The package is available online, easy to install and use.

1. The authors compared their method to existing tools which are based on two different strategies: use long reads to scaffold assemblies or error-correct long reads using short reads before the assembly step. However there is a third category based on a long-read only assembly (follow by a polishing of the consensus). The tools that belong to this category (Canu, Miniasm, Smart2Novo, as example) provide encouraging results (Castro-Wallace et al, Biorxiv, 2016 and Istace et al, BioRxiv, 2016) but are totally missing in the manuscript.

2. The authors claimed that existing methods "have not made use of the real-time sequencing potential of the MinION". It's particularly true for methods that need large computational resources, but tools like miniasm for example are able to assemble a bacterial genome in a few seconds or

minutes. Furthermore, before using npScarf one needs to produce a short-read assembly based on an illumina sequencing experiment which cannot produce data in real-time. In my opinion, a true "real-time assembler" should use nanopore data only.

3. The comparison with existing tools is not fair. Indeed, authors used 50X of illumina reads to error-correct long reads with Nanocorr and NaS, but they used the complete dataset (250X) with other tools. A lower coverage will mainly result in uncovered regions, due to the sequencing technology bias, and lead to assembly fragmentation. The authors of Nanocorr and Nas tools reported near-perfect assemblies of E. coli genome (Goodwin et al, Genome Research, 2005 and <http://www.genoscope.cns.fr/externe/nas/assemblies.html>), even with a low coverage of nanopore data. The results reported here are conflicting, is this due to a difficulty in setting the parameters?

4. I wonder if the gap filling approach is efficient, notably in tandem duplicated regions? As instance, yeast genomes contain tandemly duplicated genes. Does the number of copies observed in the npScarf assembly is consistent with the reference genome?

5. The W303 npScarf assembly contains several chimeric contigs, are they due to the presence of transposable elements? If yes, this result should be discussed in the manuscript as it may indicate the limit of the method to deal with complex genomes.

Minor revisions:

1. The method first determines the multiplicity of each contig, it could be interesting to test the method in hard conditions, for instance with single cell amplified or aneuploid genomes where the coverage per base is heterogeneous.

2. Naming of ATCC 13883 scaffolds is different from the manuscript.

3. Legend of Sup Figure 3: S2880C instead of S288C

REVIEWERS' COMMENTS:

Reviewer #1 (Remarks to the Author):

I thank the authors for their efforts to respond to my previous review. I was Reviewer #1 in the previous round, and my response to the authors' comments follow the numbering scheme they used.

1. It is great to see that the authors have modified the tone of their manuscript to drop claims about the utility of npScarf in field studies. Issue resolved.
2. This issue however remains unresolved. The point was not about processing "one" read at a time (the authors' response is that npScarf can process blocks of reads), but about the *length* of those reads processed in "real time". The manuscript has no mention of this, and certainly does not demonstrate such a use case. The claim that npScarf is a real-time algorithm is thus misleading, and the reported examples do not reflect how the nanopore technology works.
3. I continue to see a value in reporting this algorithm, though not for the use case advertised on the title.
4. The question was to prompt the authors to discuss how the tool may inform experimental design, and may be of value in go/no-go decisions to generate additional data. The authors' response is acceptable.
5. Added details on the determination of contig copy numbers are welcome. Though the methodology still lacks proper statistical basis, the response is acceptable.
6. The authors have a great idea here, even if it does not translate to a reproducible result because they decided to declare it out of scope for the current manuscript.

Reviewer #2 (Remarks to the Author):

The authors have responded to my comments well.

Reviewer #3 (Remarks to the Author):

The authors present an improved manuscript "Scaffolding and completing genome assemblies in real-time with nanopore sequencing" and my major compulsory revisions were taken into account.

Concerning the accuracy of the gap filling step and the presence of chimeric contigs, the authors considered that a comparison of their assembly with the S288C is difficult because the two strains are different. However, a reference genome generated using the PacBio technology is available for W303. Even if this assembly seems more fragmented than the one proposed in this study, the authors could use it to inspect tandemly duplicated genes as well as breakpoints (chrVI, chrXII and chr XIV).

Reviewer #1 (Remarks to the Author):

I have read Cao and co-workers' manuscript on npScarf with much interest. It describes an assembly scaffolding and finishing tool for hybrid assembly of Illumina and nanopore sequencing data. The stated motivation of the method is to provide a means for real time assembly algorithm that can be used in the field, and inform its user on when to stop a sequencing experiment.

Although the method provides competitive performance in comparison to present state-of-the-art, in my opinion it fails to deliver on its stated goals.

1. First, because it builds on assembled Illumina sequencing data, and because that step has to be performed offline, the primary use case of npScarf cannot be a field study.

We agree with the reviewer that the primary use case of npScarf is not in the field. We revised to make clearer that our method can scaffold and finish the assemblies concurrently with MinION sequencing, but the entire workflow including MiSeq sequencing cannot be performed in real-time with the current technology. We have also clarified that our primary applications are for 1. efficiently completing existing short-read genome assemblies, and 2. controlling MinION sequencing in new hybrid-assembly projects, which is substantially more expensive than short reads. [Page 2, column 1; Page 8 column 1; Page 8, column 2]

2. Second, the authors have the misconception that the MinION instrument returns its output one read at a time. Unfortunately (for the authors, but fortunately for the users) this is not the case. The instrument has many nanopores operating at the same time. So the actual data streaming happens as an asynchronous time series from multiple pores.

> npScarf actually retrieves and processes data in a block-wise fashion, i.e., a small batch of reads sequenced within a small block of time. In this sense, npScarf can make use of reads sequenced from multiple pores rather than one read at a time. The length of the time block is a runtime parameter for the algorithm. We have made this clear in the revision [Page 9, column 1]

3. Having said that, I think the study has merit if the authors are willing to reframe the target application of their tool. It performs better than the comparators on the datasets they have tested, in terms of contiguity, quality and run time. (They should also include memory comparisons in their report.)

>We have clarified the main use cases for npScarf as outlined above. We emphasise that our tool performs better than others on the tested data sets in terms of assembly contiguity and quality. We also show that our tool requires much less long read data to complete an assembly, and as such the ability to scaffold and report results on the fly will help to control MinION sequencing in completing existing short read assemblies as well as in hybrid assembly projects. [Page 2, column 1; Page 7, column 2; Page 8 column 1]

We included a brief report on the memory consumption for our method in the manuscript (<4GB which is very small memory footprint) [Page 7, column 2]. While we are convinced that our method required less memory than competitive methods, we find these pipelines have differing computational settings (nanocorr and nas distribute computation to hundred of computer nodes; scaffolding methods can only be run after the execution of Spades which can be configured for

different memory setting; and Canu and Miniasm require small memory footprint to assemble the genome, but the polishing step with Pilon consumes substantial amounts of memory), and hence a comparison is not practical. We instead include the memory consumption summary in the supplementary information [Supp. Table 1]

4. Of particular interest was, how the algorithm recovered from misassembled bridges. As the authors have demonstrated (though not explicitly stressed) as the target genome size and complexity increases, the assembly problem gets more challenging. As a result, using more data may help reduce misassemblies. This is reflected in their comparison of real time and batch assemblies of the *S. cerevisiae* dataset. My question is, can the algorithm recover from misassemblies in the batch mode. That is, if the algorithm were run in the batch mode with several partitions of the dataset (say, two or three), can the later batch runs correct for the earlier wrong decisions?

> At any one point, our algorithm can provide the most likely assembly given the available data. In real-time mode, it can correct mis-assemblies (which are mainly due to false-positive alignments at earlier stages) in light of evidence from new data. As the reviewer pointed out, using more data can generate more reliable assemblies. In the real-time mode, our algorithm evaluates and reports the assembly after one or a few reads received, whereas in the batch mode, this is done after all reads are examined. The number of reads required before a new decision is made is a parameter for the algorithm. Hence the configuration the reviewer suggested can be done by setting this number to say one half or one third of the data set. The algorithm will run in real-time mode and will correct errors from the early batch when data of a later batch provide sufficient evidence.

5. I find the way the method determines unique contigs rather simplistic, and lacks proper statistical treatment. Coverage depth of a contig depends on several factors, including its length and GC content. In general, the longer a contig, the closer its average fold coverage to the overall genome fold coverage, and I do realize that the authors are aware of that (they are considering “up to 20 of the largest contigs longer than 20 Kb”.) Of course they may provide a better justification for these thresholds, or making them run time parameters, to help address this issue to a certain extent. However it would not alleviate the problem in the way they apply an inferred cut-off across the board, as shorter contigs will have wider fluctuations in their coverage profiles.

> We provided more detailed description of the method to determine unique contigs. As the reviewer pointed out, the depth coverage estimate is more accurate with more and longer contigs. Our algorithm in fact uses an iterative approach to use the longest contigs possible so long as their depth coverage does not deviate from the estimate. In our experience, the largest 20 contigs provide sufficient ground for the estimate, i.e., the estimated coverage does not change significantly beyond 20 largest contigs. The threshold 20 Kb is another heuristic to ascertain the uniqueness of the contig when domain knowledge is available. The algorithm indeed takes this threshold as a runtime parameter, and considers a contig longer than this threshold unique without further test.

[Page 8, column 2]

6. I think, adding some logic to determine the completeness of a bacterial genome/plasmid assembly through key features is a great idea. Though it needs to be better explained how it is integrated in the assembly algorithm, including how to invoke it during a run.

>The algorithm automatically reports the statistics of the assembly (N50, total length, number of

contigs and number of circular contigs) during the scaffolding progress. Determining the completion of an assembly is left for the users depending on the domain knowledge and the applications of the experiment. For instance, the assembly of a bacterial genome can be considered completed when all contigs are circular. The users can also choose to terminate a sequencing when all plasmids are completed if plasmids are the focus of the experiment.

In our current implementation, the statistics are continuously outputted to the console or to a text format. We are working on a graphical interface for better visualisation of the assembly. We will report this in a manuscript in the near future.

Reviewer #2 (Remarks to the Author):

This is a good paper and well laid out. I am particularly impressed by (a) the variety of datasets that the authors use, including generating their own and (b) the length of accurate scaffolding this approach takes.

1. However I have one major concern which must be addressed. The authors make much of their real time aspect of scaffolding. However real time algorithms and methods are only really valid if the entire workflow can be done in real time and stopped; as this method is a hybrid method with a batched based MiSeq run, it really is not at all practical as a real time method. In theory there is a marginal saving in ONT time, but as few people directly reuse a flow cell immediately, this is not really useful.

>See response to Reviewer 1, point 1.

2. For me this should be recast as a prelude to future "all real time" methods rather than implying it could do something radical now. ie - I think it is fine to show the plots and the concept, but the authors should be up front by saying this is only really useful when the entire workflow can be stopped real time.

>See response to Reviewer 1, point 1.

Minor comments

3. Line 23 "The major advancement..." - I would say "A..." or the "the key component of the technology..."

>The phase is modified as "A key innovation of ..."

4. I prefer "less data" to "fewer data"

>We have revised as the reviewer suggested.

5. You need to provide the accession number of the MiSeq.

> We have finalised the submission of the our datasets and the ENA accessions are now provided [page 10 column 1].

Reviewer #3 (Remarks to the Author):

I read the manuscript by Cao et al., titled “Scaffolding and completing genome assemblies in real-time with nanopore sequencing” with great interest. The authors describe a method dedicated to de novo genome assembly of small genomes by combining data generated using the Oxford Nanopore and the Illumina sequencing technologies. Moreover, authors take advantage of the “real-time” capability of the MinION device, and they propose to improve an existing assembly in real-time. The manuscript particularly well described the impact of sequencing coverage on the assembly contiguity. The npScarf software is of broad interest for the nanopore users. The package is available online, easy to install and use.

1. The authors compared their method to existing tools which are based on two different strategies: use long reads to scaffold assemblies or error-correct long reads using short reads before the assembly step. However there is a third category based on a long-read only assembly (follow by a polishing of the consensus). The tools that belong to this category (Canu, Miniasm, Smart2Novo, as example) provide encouraging results (Castro-Wallace et al, Biorxiv, 2016 and Istace et al, BioRxiv, 2016) but are totally missing in the manuscript.

> We have now included this category (de novo assembly + polish with short reads) in our manuscript [Page 1, column 2; Page 8 column 1]. We also included Canu and Miniasm in our comparison in this revision. As these methods are de novo assembly methods, they require significantly more long read data than our method; evidently, they failed to produce a decent assembly for the bacterial data sets (low coverage). On the yeast data set, they used twice as much data as npScarf, and yet their assemblies are less complete and less contiguous [Page 5; Page 7 column 2; Table 3].

The two references mentioned by the reviewer were posted on Biorxiv after the submission of our paper. We now included these in this revision. [Page 2, column 1].

2. The authors claimed that existing methods “have not made use of the real-time sequencing potential of the MinION”. It’s particularly true for methods that need large computational resources, but tools like miniasm for example are able to assemble a bacterial genome in a few seconds or minutes. Furthermore, before using npScarf one needs to produce a short-read assembly based on an illumina sequencing experiment which cannot produce data in real-time. In my opinion, a true “real-time assembler” should use nanopore data only.

> As in the response to Reviewer 1, point 1, we agree that a true “real-time” assembler” should use only nanopore data. In that sense, the npScarf pipeline is not real-time as it requires the prior short read assembly. We have reframed our algorithm as a real-time scaffolder that can be used to control resources during MinION sequencing.

3. The comparison with existing tools is not fair. Indeed, authors used 50X of illumina reads to

error-correct long reads with Nanocorr and NaS, but they used the complete dataset (250X) with other tools. A lower coverage will mainly result in uncovered regions, due to the sequencing technology bias, and lead to assembly fragmentation. The authors of Nanocorr and Nas tools reported near-perfect assemblies of E. coli genome (Goodwin et al, Genome Research, 2005 and <http://www.genoscope.cns.fr/externe/nas/assemblies.html>), even with a low coverage of nanopore data. The results reported here are conflicting, is this due to a difficulty in setting the parameters?

> We used only 50X of Illumina reads for error-correction following their recommendations (the accuracy of these methods does not increase with more than 30X of Illumina data). Even with only 50X coverage of Illumina, these methods took very long time (over 7000 CPU-hours on the yeast data set), and the running time grows (linearly) with the size of the Illumina data.

The E.coli data set from which most tool reported near perfect assemblies, was from three flowcells reported by Quick et al 2015. The total coverage from the three flowcells is 147-fold. We used data from only one flowcell (the R7.3 flowcell) which gave 67-fold coverage.

The coverage reported in the web site was the coverage of NaS assembly reads, that is reads that have been corrected. NaS used only 2D reads from the 147-fold coverage of the mentioned E. coli dataset (which make up only 25% of the dataset). These reads are then corrected, and only reads that can be corrected are selected and reported. These coverage statistics do not reflect the amount of raw nanopore data. In our manuscript, we reported the coverage based on all raw nanopore data (both 1D and 2D).

The assembly results from Nanocorr and Nas are very sensitive to the parameters (during assembly with Celera Assembler). We used the multiple specification files used by NaS and Nanocorr for each data set, and reported the best results. Our results from running these two methods on the yeast data set (the same data) were consistent with what reported in their publication. The results on the E. coli assemblies were different because we used only a subset of their data set, as described above.

4. I wonder if the gap filling approach is efficient, notably in tandem duplicated regions? As instance, yeast genomes contain tandemly duplicated genes. Does the number of copies observed in the npScarf assembly is consistent with the reference genome?

> npScarf fills in the gaps by aligning the short contigs to the bridges in the gaps created by long reads. One thing to note is that the size of the gap between non-repetitive contigs (and hence the number of repeat units) is determined by the nanopore reads. It is difficult to draw firm conclusions on accuracy of repeat unit typing from the yeast sample (W303) as the reference sample is not the same as the sequenced strain (S228C). We feel that a comparison of the repeat length typing accuracy of npScarf to other assemblers is beyond the scope of the current paper.

5. The W303 npScarf assembly contains several chimeric contigs, are they due to the presence of transposable elements? If yes, this result should be discussed in the manuscript as it may indicate the limit of the method to deal with complex genomes.

>The reference strain (S228C) is different to the strain from which the sequence data was

obtained (W303) and hence we cannot easily distinguish rearrangements between these two strains and genuine assembly errors. However, we can compare the number of apparent assembly errors between npScarf and other assemblers, and we see from this the npScarf is highly competitive. We have included dot plots of the Canu and Miniasm assemblers for comparative purposes (Supp fig 3&4). We have inserted the following text to acknowledge the challenges posed by interspersed repeats to npScarf as well as other assemblers (page 7 col 1)

"We found these mis-assemblies were due to the presence of interspersed repeat elements which are known for being problematic in assembly analysis~\cite{TreangenS2012}. The assemblies produced by Canu and Miniasm also presented several mis-assemblies fusing different chromosomes together, emphasising the challenges posed by interspersed repeats in assembling complex genomes (See Supplementary Figures 4 and 5)."

Minor revisions:

1. The method first determines the multiplicity of each contig, it could be interesting to test the method in hard conditions, for instance with single cell amplified or aneuploid genomes where the coverage per base is heterogeneous.

This is an interesting suggestion. We will investigate this in further research.

2. Naming of ATCC 13883 scaffolds is different from the manuscript.

3. Legend of Sup Figure 3: S2880C instead of S288C

> We have revised the manuscript to make sure namings are correct and consistent.

Reviewer #1 (Remarks to the Author):

I thank the authors for their efforts to respond to my previous review. I was Reviewer #1 in the previous round, and my response to the authors' comments follow the numbering scheme they used.

1. It is great to see that the authors have modified the tone of their manuscript to drop claims about the utility of npScarf in field studies. Issue resolved.
2. This issue however remains unresolved. The point was not about processing "one" read at a time (the authors' response is that npScarf can process blocks of reads), but about the *length* of those reads processed in "real time". The manuscript has no mention of this, and certainly does not demonstrate such a use case. The claim that npScarf is a real-time algorithm is thus misleading, and the reported examples do not reflect how the nanopore technology works.

Response:

To avoid confusion, we have added the following sentence in the discussion:

“Our pipeline processes a sequence read within minutes of it finishing traversing the pore, rather than as the read is actually passing through the pore, and as such is real-time at the temporal resolution of minutes, but not at the millisecond level required to update with the addition of each base.”

And another in the introduction

“In effect, npScarf can fully utilise a sequence read within minutes of it being generated. Furthermore, it continuously reports assembly quality during the experiment so that users can terminate the sequencing when an assembly of sufficient quality and completeness is obtained.”

3. I continue to see a value in reporting this algorithm, though not for the use case advertised on the title.

Response: We have further clarified the use case in the abstract and paper body, and made clear in the discussion and introduction that our use of realtime relates to minutes, rather than milliseconds.

4. The question was to prompt the authors to discuss how the tool may inform experimental design, and may be of value in go/no-go decisions to generate additional data. The authors' response is acceptable.
5. Added details on the determination of contig copy numbers are welcome. Though the methodology still lacks proper statistical basis, the response is acceptable.
6. The authors have a great idea here, even if it does not translate to a reproducible result because they decided to declare it out of scope for the current manuscript.

Reviewer #2 (Remarks to the Author):

The authors have responded to my comments well.

Reviewer #3 (Remarks to the Author):

The authors present an improved manuscript “Scaffolding and completing genome assemblies in real-time with nanopore sequencing” and my major compulsory revisions were taken into account.

Concerning the accuracy of the gap filling step and the presence of chimeric contigs, the authors considered that a comparison of their assembly with the S288C is difficult because the two strains are different. However, a reference genome generated using the PacBio technology is available for W303. Even if this assembly seems more fragmented than the one proposed in this study, the authors could use it to inspect tandemly duplicated genes as well as breakpoints (chrVI, chrXII and chr XIV).

Response: As noted by the reviewer this assembly is quite fragmented and as such we decided not to use it.